# Sufficient and Necessary Explanations
# (and What Lies in Between)

Beepul Bharti[1], Paul Yi[2], Jeremias Sulam[1]
[1]Johns Hopkins University, [2]St. Jude Children's Research Hospital
`bbharti1@jh.edu, paul.yi@stjude.org, jsulam@jhu.edu`

As complex machine learning models continue to be used in high-stakes decision settings, understanding their predictions is crucial. Post-hoc explanation methods aim to identify which features of an input $\mathbf{x}$ are important to a model's prediction $f(\mathbf{x})$. However, explanations often vary between methods and lack clarity, limiting the information we can draw from them. To address this, we formalize two precise concepts—*sufficiency* and *necessity*—to quantify how features contribute to a model's prediction. We demonstrate that, although intuitive and simple, these two types of explanations may fail to fully reveal which features a model deems important. To overcome this, we propose and study a unified notion of importance that spans the entire sufficiency-necessity axis. Our unified notion, we show, has strong ties to notions of importance based on conditional independence and Shapley values. Lastly, through various experiments, we quantify the sufficiency and necessity of popular post-hoc explanation methods. Furthermore, we show that generating explanations along the sufficiency-necessity axis can uncover important features that may otherwise be missed, providing new insights into feature importance.

## 1. Introduction

Over recent years, modern machine learning (ML) models, mostly deep learning-based, have achieved impressive results across several complex domains. Models can now solve difficult problems in computer vision, perform accurate text and sentiment analysis, predict the three-dimensional conformation of proteins, and more [1, 2]. Despite their success, the rapid integration of these models into society requires caution [3]. Modern ML systems are black-boxes, comprised of millions of parameters and non-linearities that obscure their prediction-making mechanisms from everyone. This lack of clarity raises concerns about explainability, transparency, and accountability [4, 5]. Thus, understanding how these models work is essential for their safe deployment.

The lack of explainability has spurred research efforts in eXplainable AI (XAI). One major focus is on developing post-hoc methods to explain black-box model predictions, especially at a *local* level. For a model $f$ and input $\mathbf{x}$, these methods aim to identify which features in $\mathbf{x}$ are *important* for the prediction, $f(\mathbf{x})$. They do so by estimating a notion of importance for each feature (or groups), which allows for a ranking of importance. Popular methods include CAM [6], LIME [7], gradient-based approaches [8–10], rate-distortion techniques [11], Shapley value-based explanations [12–14], perturbation-based methods [15–17], among others [18–22]. Unfortunately, many of these approaches lack rigor, as the meaning of their computed scores is often ambiguous. For example, it's not always clear what large or negative gradients signify or what high Shapley values reveal about feature importance. To address these concerns, other work has focused on methods based on propositional logic [23–26], conditional hypothesis testing [27, 28], among formal notions. While these methods are a step towards rigor, they have drawbacks, including reliance on complex reasoners and limited ability to communicate their results in an understandable way to human decision-makers.

In this work, we advance XAI research by providing formal mathematical definitions of *sufficient* and *necessary* features for explaining complex ML models. First, we illustrate how, although informative, sufficient and necessary explanations offer incomplete insights into feature importance. To address this, we propose and study a more general unified framework for explaining models. Finally, we

Second Conference on Parsimony and Learning (CPAL 2025).

offer two novel perspectives on our framework through the lens of conditional independence and Shapley values, and crucially, show how it can reveal new insights into feature importance.

## 1.1. Summary of our Contributions

We propose and study two approaches, sufficiency, and necessity, which evaluate the contribution of a set of features in $\mathbf{x}$ toward a model prediction $f(\mathbf{x})$. A sufficient set preserves the model's output, while a necessary set, when removed, renders the output uninformative. Although the two concepts appear complementary, their precise relationship remains unclear. How similar are sufficient and necessary subsets? How different? To address these questions, we study the two concepts and propose a *unification* of both. Our contributions are summarized as follows:

1. We formalize precise mathematical definitions of sufficient and necessary features for model predictions that are related but complementary to those in previous works.

2. We propose a unified approach that combines sufficiency and necessity, exploring when and how they align or differ. Additionally, we motivate its utility by highlighting its connections to conditional independence and Shapley values, a game-theoretic measure of feature importance.

3. Through experiments of increasing complexity, we demonstrate how a unified perspective uncovers new, significant, and more comprehensive insights into feature importance.

## 2. Sufficiency and Necessity

**Notation & Setting.** We use boldface uppercase letters to denote random vectors (e.g., $\mathbf{X}$) and lowercase for their values (e.g., $\mathbf{x}$). For a subset $S \subseteq [d] := \{1, \ldots, d\}$, we denote its cardinality by $|S|$ and its complement $S^c = [d] \setminus S$. Subscripts index features; e.g., $\mathbf{x}_S$ represents $\mathbf{x}$ restricted to the entries indexed by $S$. We consider a supervised learning setting with an unknown distribution $\mathcal{D}$ over features $\mathcal{X} \subseteq \mathbb{R}^d$ and labels $\mathcal{Y} \subseteq \mathbb{R}$ and assume access to a model $f : \mathcal{X} \mapsto \mathcal{Y}$ trained on samples from $\mathcal{D}$. For an input $\mathbf{x} = (x_1, \ldots, x_d) \in \mathbb{R}^d$, the goal is to identify the important features in $\mathbf{x}$ for the prediction $f(\mathbf{x})$. To define importance, we use the average restricted prediction [29, 30]

$$f_S(\mathbf{x}) = \mathop{\mathbb{E}}_{\mathbf{X}_{S^c} \sim \mathcal{V}_{S^c}} [f(\mathbf{x}_S, \mathbf{X}_{S^c})] \tag{1}$$

where $\mathbf{x}_S$ is fixed and $\mathbf{X}_{S^c}$ is a random vector drawn from an arbitrary reference distribution $\mathcal{V}_{S^c}$ (which may or may not depend on $S^c$). Two common choices for $\mathcal{V}_{S^c}$ are the marginal $p(\mathbf{X}_{S^c})$ and conditional distribution $p(\mathbf{X}_{S^c} \mid \mathbf{x}_S)$. With $f_S(\mathbf{x})$ we can query $f$, which only takes inputs in $\mathbb{R}^d$, and analyze its behavior when sets of features are retained or removed.

**Definitions.** We now present our proposed definitions of sufficiency and necessity. At a high level, these definitions were formalized to align with the following guiding principles:

**P1**. $S$ is sufficient if it is enough to generate the original prediction, i.e. $f_S(\mathbf{x}) \approx f(\mathbf{x})$.

**P2**. $S$ is necessary if we cannot generate the original prediction without it, i.e. $f_{S^c}(\mathbf{x}) \not\approx f(\mathbf{x})$.

**P3**. The set $S = [d]$ should be maximally sufficient and necessary for $f(\mathbf{x})$.

The principles P1 and P2 are natural and agree with the logical notions of sufficiency and necessity. Furthermore, because the full set of features provides all the information needed to make the prediction $f(\mathbf{x})$, it should thus be regarded as maximally sufficient and necessary (P3). With these principles laid out, we now formally define sufficiency and necessity.

**Definition 2.1** (Sufficiency). *Let $\epsilon \geq 0$ and let $\rho : \mathbb{R} \times \mathbb{R} \mapsto \mathbb{R}$ be a metric on $\mathbb{R}$. A subset $S \subseteq [d]$ is $\epsilon$-sufficient with respect to a distribution $\mathcal{V}$ for $f$ at $\mathbf{x}$ if*

$$\Delta_{\mathcal{V}}^{suf}(S, f, \mathbf{x}) \triangleq \rho(f(\mathbf{x}), f_S(\mathbf{x})) \leq \epsilon. \tag{2}$$

*Furthermore, $S$ is $\epsilon$-super sufficient if all supersets $\widetilde{S} \supseteq S$ are $\epsilon$-sufficient.*

This notion of sufficiency is straightforward and aligns with P1. A subset $S$ is $\epsilon$-sufficient with respect to reference distribution $\mathcal{V}$ if, with $\mathbf{x}_S$ fixed, the average restricted prediction $f_S(\mathbf{x})$ is within

$\epsilon$ from the original $f(\mathbf{x})$. Furthermore, $S$ is $\epsilon$-super sufficient if $\rho(f(\mathbf{x}), f_S(\mathbf{x})) \leq \epsilon$ and, $\forall \widetilde{S} \supseteq S$, $\rho(f(\mathbf{x}), f_{\widetilde{S}}(\mathbf{x})) \leq \epsilon$. Namely, including more features in $S$ keeps $f_S(\mathbf{x})$ $\epsilon$ close to $f(\mathbf{x})$. Note this definition aligns with P3, since the set $S = [d]$ is 0-sufficient (maximally sufficient). To find a small sufficient subset $S$ of small cardinality $\tau > 0$, we can solve the following optimization problem:

$$\underset{S \subseteq [d]}{\arg\min} \ \Delta_{\mathcal{V}}^{\mathsf{suf}}(S, f, \mathbf{x}) \ \text{subject to} \ |S| \leq \tau \qquad (\mathsf{P_{suf}})$$

We will refer to this problem as the *sufficiency problem*, or ($\mathsf{P_{suf}}$). Using analogous ideas, we also define necessity and formulate an optimization problem to find small necessary subsets.

**Definition 2.2** (Necessity). *Let $\epsilon \geq 0$ and denote $\rho : \mathbb{R} \times \mathbb{R} \mapsto \mathbb{R}$ to be metric on $\mathbb{R}$. A subset $S \subseteq [d]$ is $\epsilon$-necessary with respect to a distribution $\mathcal{V}$ for $f$ at $\mathbf{x}$ if*

$$\Delta_{\mathcal{V}}^{\mathsf{nec}}(S, f, \mathbf{x}) \triangleq \rho(f_{S^c}(\mathbf{x}), f_{\emptyset}(\mathbf{x})) \leq \epsilon. \qquad (3)$$

*Furthermore, $S$ is $\epsilon$-super necessary if all supersets $\widetilde{S} \supseteq S$ are $\epsilon$-necessary.*

Here, a subset $S$ is $\epsilon$-necessary if marginalizing out the features in $S$ with respect to $\mathcal{V}_S$, results in an average restricted prediction $f_{S^c}(\mathbf{x})$ that is $\epsilon$ close to $f_{\emptyset}(\mathbf{x})$ – the average baseline prediction of $f$ over $\mathcal{V}_{[d]}$. Furthermore, $S$ is $\epsilon$-super necessary if $\rho(f_S(\mathbf{x}), f(\mathbf{x})) \leq \epsilon$ and all super sets of $S$ are $\epsilon$-necessary. Note, our definition of differs from alternatives [31, 32] which state that $S$ is necessary if $\rho(f(\mathbf{x}), f_{S^c}(\mathbf{x})) \geq \gamma$ for some $\gamma > 0$. Our notion is more general in that it implies this condition. Intuitively, if $f_{\emptyset}(\mathbf{x})$ and $f(\mathbf{x})$ differ, and $f_{S^c}(\mathbf{x})$ is close to $f_{\emptyset}(\mathbf{x})$, then $f_{S^c}(\mathbf{x})$ and $f(\mathbf{x})$ will also differ. Furthermore, for $S = [d]$, we have $\Delta_{\mathcal{V}}^{\mathsf{nec}}(S, f, \mathbf{x}) \triangleq \rho(f_{\emptyset}(\mathbf{x}), f_{\emptyset}(\mathbf{x})) = 0$, indicating that $S = [d]$ is 0-necessary (maximally necessary) as desired. To identify a necessary subset $S$ of small cardinality $\tau > 0$, one can solve the following problem, which we refer to as the *necessity* problem or ($\mathsf{P_{nec}}$).

$$\underset{S \subseteq [d]}{\arg\min} \ \Delta_{\mathcal{V}}^{\mathsf{nec}}(S, f, \mathbf{x}) \ \text{subject to} \ |S| \leq \tau \qquad (\mathsf{P_{nec}})$$

Having presented our definitions, we now discuss related works before presenting our main results.

## 3. Related Work

Notions of sufficiency, necessity, their duality and connections with other feature attribution methods have been studied to varying degrees. We comment on the main related works in this section.

**Sufficiency.** The notion of sufficient features has gained significant attention in recent research. Shih et al. [26] explore a symbolic approach to explain Bayesian network classifiers and introduce prime implicant explanations, which are minimal subsets $S$ that make features in the complement irrelevant to the prediction $f(\mathbf{x})$. For models represented by a finite set of first-order logic (FOL) sentences, Ignatiev et al. [23] refer to prime implicants as abductive explanations (AXp's). For classifiers defined by propositional formulas and inputs with discrete features, Darwiche and Hirth [24] refer to prime implicants as sufficient reasons and define a complete reason to be the disjunction of all sufficient reasons. They present efficient algorithms, leveraging Boolean circuits, to compute sufficient and complete reasons and demonstrate their use in identifying classifier dependence on protected features that should not inform decisions. For more complex models, Ribeiro et al. [22] propose high-precision probabilistic explanations called anchors, which represent local, sufficient conditions. For $\mathbf{x}$ positively classified by $f$, Wang et al. [21] propose a greedy approach to solve ($\mathsf{P_{suf}}$), I Amoukou and Brunel [33] extend this work to regression settings using tree-based models, and Fong and Vedaldi [15] introduce the preservation method which relaxes $S$ to $[0, 1]^d$.

**Necessity.** There has also been significant focus on identifying necessary features – those that, when altered, lead to a change in the prediction $f(\mathbf{x})$. For models expressible by FOL sentences, Ignatiev et al. [34] define prime implicates as the minimal subsets that when changed, modify the prediction and relate these to adversarial examples. For Boolean models and samples $\mathbf{x}$ with discrete features, Ignatiev et al. [23] and [24] refer to prime implicates as contrastive explanations (CXp's) and necessary reasons, respectively. Beyond boolean functions, for $\mathbf{x}$ positively classified by a classifier $f$, Fong et al. [16] relax $S$ to $[0, 1]^d$ and propose the deletion method to approximately solve ($\mathsf{P_{nec}}$).

**Duality Between Sufficiency and Necessity.** Dabkowski and Gal [17] characterize the preservation and deletion methods as discovering the *smallest sufficient* and *destroying region* (SSR and SDR). They propose combining the two but do not explore how solutions to this approach may differ from individual SSR and SDR solutions. Ignatiev et al. [23] show that AXp's and CXp's are minimal hitting sets of another by using a hitting set duality result between minimal unsatisfiable and correction subsets. The result enables the identification of AXp's from CXp's and vice versa.

**Sufficiency, Necessity, and General Feature Attribution Methods.** Precise connections between sufficiency, necessity, and other popular feature attribution methods (such as Shapley values [12, 29, 35]) remains unclear. To our knowledge, Covert et al. [36] provide the only work examining these approaches [15–17] in the context of general removal-based methods, i.e., methods that remove certain input features to evaluate different notions of importance. The work of Watson et al. [37] is also relevant to our work, as it formalizes a connection between notions of sufficiency and Shapley values. With the specific payoff function defined as $v(S) = \mathbb{E}[f(\mathbf{x}_S, \mathbf{X}_{S^c})]$, they show how each summand in the Shapley value measures the sufficiency of feature $i$ to a particular subset.

# 4. Unifying Sufficiency and Necessity

Given a model $f$ and sample $\mathbf{x}$, we can identify a small set of important features $S$ by solving either $(\mathrm{P_{suf}})$ or $(\mathrm{P_{nec}})$. While both methods are popular [11, 15, 19, 38]. identifying small sufficient or necessary subsets may not provide a complete picture of how $f$ uses $\mathbf{x}$ to make a prediction. To see why, consider the following scenario: for a fixed $\tau > 0$, let $S^*$ be a $\epsilon$-sufficient solution to $(\mathrm{P_{suf}})$, so that $\Delta_{\mathcal{V}}^{\mathsf{suf}}(S, f, \mathbf{x}) \leq \epsilon$. While $S^*$ is $\epsilon$-sufficient, it can also be true that $\Delta_{\mathcal{V}}^{\mathsf{nec}}(S, f, \mathbf{x}) > \epsilon$ indicating $S^*$ is **not** $\epsilon$-necessary: indeed, this can simply happen when its complement, $S^{c*}$, contains important features. This scenario raises two questions: 1) How different are sufficient and necessary features? 2) How does varying the levels of sufficiency and necessity affect the optimal set of important features?

To answer these important questions (and avoid the scenario above) we propose studying a unification of $(\mathrm{P_{suf}})$ and $(\mathrm{P_{nec}})$. Consider $\Delta_{\mathcal{V}}^{\mathsf{uni}}(S, f, \mathbf{x}, \alpha) = \alpha \cdot \Delta_{\mathcal{V}}^{\mathsf{suf}}(S, f, \mathbf{x}) + (1 - \alpha) \cdot \Delta_{\mathcal{V}}^{\mathsf{nec}}(S, f, \mathbf{x})$, a convex combination of $\Delta_{\mathcal{V}}^{\mathsf{suf}}(S, f, \mathbf{x})$ and $\Delta_{\mathcal{V}}^{\mathsf{nec}}(S, f, \mathbf{x})$, where $\alpha \in [0, 1]$ controls the extent to which $S$ is sufficient vs. necessary. Our *unified problem*, $(\mathrm{P_{uni}})$, can be expressed as:

$$\underset{S \subseteq [d]}{\arg\min} \ \Delta_{\mathcal{V}}^{\mathsf{uni}}(S, f, \mathbf{x}, \alpha) \ \text{ subject to } |S| \leq \tau \qquad (\mathrm{P_{uni}})$$

When $\alpha$ is 1 or 0, $\Delta_{\mathcal{V}}^{\mathsf{uni}}(S, f, \mathbf{x}, \alpha)$ reduces to $\Delta_{\mathcal{V}}^{\mathsf{suf}}(S, f, \mathbf{x})$ or $\Delta_{\mathcal{V}}^{\mathsf{nec}}(S, f, \mathbf{x})$, respectively. In these extreme cases, $S$ is only sufficient or necessary. In the remainder of this work we will analyze $(\mathrm{P_{uni}})$, characterize its solutions, and provide different interpretations of what properties the solutions have through the lens of conditional independence and game theory. In the experimental section, we will show that solutions to $(\mathrm{P_{uni}})$ provide insights that neither $(\mathrm{P_{suf}})$ nor $(\mathrm{P_{nec}})$ offer.

## 4.1. Solutions to the Unified Problem

We begin with a simple lemma that demonstrates why $(\mathrm{P_{uni}})$ enforces both sufficiency and necessity.

**Lemma 4.1.** *Let $\alpha \in (0, 1)$. For $\tau > 0$, denote $S^*$ to be a solution to $(\mathrm{P_{uni}})$ for which $\Delta_{\mathcal{V}}^{uni}(S, f, \mathbf{x}, \alpha) = \epsilon$. Then, $S^*$ is $\frac{\epsilon}{\alpha}$-sufficient and $\frac{\epsilon}{1-\alpha}$-necessary.*

The proof of this result, and all others, is included Appendix A.1. This result illustrates that solutions to $(\mathrm{P_{uni}})$ satisfy varying definitions of sufficiency and necessity. Furthermore, as $\alpha$ increases from 0 to 1, the solution shifts from being highly necessary to highly sufficient. In the following results, we will show *when* and *how* solutions to $(\mathrm{P_{uni}})$ are similar (and different) to those of $(\mathrm{P_{suf}})$ and $(\mathrm{P_{nec}})$. To start, we present the following lemma, which will be useful in subsequent results.

**Lemma 4.2.** *For $0 \leq \epsilon < \frac{\rho(f(\mathbf{x}), f_\emptyset(\mathbf{x}))}{2}$, denote $S_{suf}^*$ and $S_{nec}^*$ to be $\epsilon$-sufficient and $\epsilon$-necessary sets. Then, if $S_{suf}^*$ is $\epsilon$-super sufficient or $S_{nec}^*$ is $\epsilon$-super necessary, we have $S_{suf}^* \cap S_{nec}^* \neq \emptyset$.*

This lemma demonstrates that, given $\epsilon$-sufficient and necessary sets $S_{\mathsf{suf}}^*$ and $S_{\mathsf{nec}}^*$, if either additionally satisfies the stronger notions of super sufficiency or necessity, they must share some features. This proves useful in characterizing a solution to $(\mathrm{P_{uni}})$, which we now do in the following theorem.

**Theorem 4.1.** *Let $\tau_1, \tau_2 > 0$ and $0 \leq \epsilon < \frac{1}{2} \cdot \rho(f(\mathbf{x}), f_\emptyset(\mathbf{x}))$. Denote $S^*_{suf}$ and $S^*_{nec}$ to be $\epsilon$-super sufficient and $\epsilon$-super necessary solutions to $(\mathrm{P_{suf}})$ and $(\mathrm{P_{nec}})$, respectively, such that $|S^*_{suf}| = \tau_1$ and $|S^*_{nec}| = \tau_2$. Then, there exists a set $S^*$ such that*

$$\Delta^{uni}_{\mathcal{V}}(S^*, f, \mathbf{x}, \alpha) \leq \epsilon \quad and \quad \max(\tau_1, \tau_2) \leq |S^*| < \tau_1 + \tau_2. \tag{4}$$

*Furthermore, if $S^*_{suf} \subseteq S^*_{nec}$ or $S^*_{nec} \subseteq S^*_{suf}$, then $S^* = S^*_{nec}$ or $S^* = S^*_{suf}$, respectively.*

This result demonstrates that when there are $\epsilon$-super sufficient and $\epsilon$-super necessary solutions to $(\mathrm{P_{suf}})$ and $(\mathrm{P_{nec}})$, then one can identify a set $S^*$ with small $\Delta^{uni}$. As an example, consider features that are $\epsilon$-super sufficient, $S^*_{suf}$. If we have domain knowledge that $S^*_{suf} \subseteq S^*_{nec}$, and $S^*_{nec}$ is $\epsilon$-super necessary, then $S^*_{nec}$ will have a small $\Delta^{uni}$. Conversely, if we know that $S^*_{suf}$ is $\epsilon$-super necessary along with being a subset of $\epsilon$-super sufficient set $S^*_{suf}$, then $S^*_{suf}$ will have a small $\Delta^{uni}$.

# 5. Two Perspectives of the Unified Approach

In the previous section, we characterized solutions to $(\mathrm{P_{uni}})$ and their connections to those of $(\mathrm{P_{suf}})$ and $(\mathrm{P_{nec}})$. To further motivate and the unified approach, we now offer two alternative perspectives of our framework through the lens of conditional independence and Shapley values.

## 5.1. A Conditional Independence Perspective

Here we demonstrate how sufficiency, necessity, and their unification, can be understood as conditional independence relations between features $\mathbf{X}$ and label $Y$.

**Corollary 5.1.** *Suppose $\forall S \subseteq [d], \mathcal{V}_S = p(\mathbf{X}_S | \mathbf{X}_{S^c} = \mathbf{x}_{S^c})$. Let $\alpha \in (0, 1), \epsilon \geq 0$, and denote $\rho : \mathbb{R} \times \mathbb{R} \mapsto \mathbb{R}$ to be a metric. Furthermore, for $\tau > 0$ and $f(\mathbf{X}) = \mathbb{E}[Y \mid \mathbf{X}]$, let $S^*$ be a solution to $(\mathrm{P_{uni}})$ such that $\Delta^{uni}_{\mathcal{V}}(S, f, \mathbf{x}, \alpha) = \epsilon$. Then, $S^*$ satisfies the follow conditional independencies,*

$$\rho\left(\mathbb{E}[Y \mid \mathbf{x}], \mathbb{E}[Y \mid \mathbf{X}_{S^*} = \mathbf{x}_{S^*}]\right) \leq \frac{\epsilon}{\alpha} \quad and \quad \rho\left(\mathbb{E}[Y \mid \mathbf{X}_{S^*_c} = \mathbf{x}_{S^*_c}], \mathbb{E}[Y]\right) \leq \frac{\epsilon}{1 - \alpha}. \tag{5}$$

The assumption here is that $f_S(\mathbf{x})$ is evaluated using the conditional distribution $p(\mathbf{X}_{S^c} \mid \mathbf{X}_S = \mathbf{x}_S)$ as the reference $\mathcal{V}_S$. Given the recent advancements in generative models [39–41], this assumption is (approximately) reasonable in many settings, as we will demonstrate in our experiments. For this choice of $\mathcal{V}_S$ and model $f(\mathbf{X}) = \mathbb{E}[Y \mid \mathbf{X}]$, the result shows that minimizing $(\mathrm{P_{uni}})$ identifies an $S^*$ that approximately satisfies two conditional independence properties. First, $S^*$ is sufficient as conditioning on $S^*$ leaves the complement $S^{c*}$ with minimal additional information about $Y$. Second, $S^*$ is necessary because when we only rely on the complement $S^{c*}$, the information gained about $Y$ is minimal and similar to $\mathbb{E}[Y = 1]$.

## 5.2. A Shapley Value Perspective

In the previous section, we detailed the conditional independence relations being optimized for when demanding sufficiency, necessity, or both. We now present an arguably less intuitive result that shows that solving $(\mathrm{P_{uni}})$ is equivalent to maximizing the lower bound of the Shapley value. Before presenting our result, we provide a brief background on this game-theoretic quantity.

**Shapley Values.** Shapley values use game theory to measure the importance of players in a game. Let the tuple $([n], v)$ represent a cooperative game with players $[n] = \{1, 2, \ldots, n\}$ and denote a characteristic function $v(S) : \mathcal{P}([n]) \to \mathbb{R}$, The Shapley value [35] for player $j$ in the game $([n], v)$ is $\phi^{shap}_j([n], v) = \sum_{S \subseteq [n] \setminus \{j\}} w_S \cdot [v(S \cup \{j\}) - v(S)]$ where $w_S = \frac{|S|!(n - |S| - 1)!}{n!}$. In the context of XAI, Shapley values are widely used to measure local feature importance by treating input features as players in a game [12, 13, 29, 42]. Given a sample $\mathbf{x}$ and a model $f$, the importance of $x_j$ to the prediction $f(\mathbf{x})$ is measured by computing $\phi^{shap}_j$ for a game $([d], v)$, where $v(S)$ quantifies how the features in $S$ contribute to $f(\mathbf{x})$. Different choices of $v(S)$ can be found in [29, 43, 44]. Although computing $\phi^{shap}_j$ is computationally intractable, several practical methods for estimation have been developed [13, 30, 45, 46]. While Shapley values are popular across various domains [47–49], few works, aside from Watson et al. [37], explore their connections to sufficiency and necessity.

With this background, we now present our result. Recall solving ($P_{uni}$) finds a small subset $S$ with low $\Delta_{\mathcal{V}}^{uni}(S, f, \mathbf{x}, \alpha)$. Notice that ($P_{uni}$) naturally *partitions* the features into two sets, $S$ and $S^c$. In the following theorem we demonstrate that finding a small $S$ with minimal $\Delta_{\mathcal{V}}^{uni}(S, f, \mathbf{x}, \alpha)$ is equivalent to maximizing a lower bound on the Shapley value in a two player game.

**Theorem 5.1.** *Consider an input $\mathbf{x}$ for which $f(\mathbf{x}) \neq f_\emptyset(\mathbf{x})$. Denote by $\Lambda_d = \{S, S^c\}$ the partition of $[d] = \{1, 2, \ldots, d\}$, and define the characteristic function to be $v(S) = -\rho(f(\mathbf{x}), f_S(\mathbf{x}))$. Then,*

$$\phi_S^{shap}(\Lambda_d, v) \geq \rho(f(\mathbf{x}), f_\emptyset(\mathbf{x})) - \Delta_{\mathcal{V}}^{uni}(S, f, \mathbf{x}, \alpha). \tag{6}$$

This result motivates minimizing $\Delta^{uni}$ through a game-theoretic interpretation. The tuple $(\Lambda_d, v)$ defines a game, and with $2^{d-1}$ ways to partition $[d]$, there are $2^{d-1}$ games, with the inequality holding for all of them. Thus, Theorem 5.1 shows that finding the $S$ with minimal $\Delta^{uni}$ is equivalent to identifying the the game (i.e. partition) where $S$ has the largest lower bound on its Shapley value.

# 6. Solving the Unified Problem

Before presenting our results, we briefly discuss approaches to solving ($P_{uni}$). While the problem is NP-hard, exact solutions can be efficiently computed or approximated using tractable relaxations in certain settings [11, 16, 50]. We provide an overview here and defer details to Appendix A.3.

**Exhaustive Search.** When the feature space dimension $d$ or the choice of $\tau \in \mathbb{Z}_{>0}$ is small, an exhaustive search can compute exact solutions to ($P_{uni}$) by evaluating $\Delta_{\mathcal{V}}^{uni}(S, f, \mathbf{x}, \alpha)$ for all $\binom{d}{\tau}$ subsets $S$ of cardinality $\tau$ and selecting the minimizer.

**Instance-wise Optimization.** When $d$ is large, rendering ($P_{uni}$) intractable, one can generate approximate solutions by solving the relaxed problem[1]

$$\arg\min_{S \subseteq [0,1]^d} \Delta_{\mathcal{V}}^{uni}(S, f, \mathbf{x}, \alpha) + \lambda_1 \cdot ||S||_1 + \lambda_{TV} \cdot ||S||_{TV}. \tag{7}$$

This approach is common in computer vision and natural language problems [11, 16, 50, 51] to generate instance-specific solutions.

**Parametric Model Approach.** Another approach we to generate solutions to ($P_{uni}$) is to learn models $g_\theta : \mathcal{X} \mapsto [0, 1]^d$ that (approximately) solve the following optimization problem:

$$\arg\min_{\theta \in \Theta} \mathbb{E}_{\mathbf{X} \sim \mathcal{D}_{\mathcal{X}}} \left[ \Delta_{\mathcal{V}}^{uni}(g_\theta(\mathbf{X}), f, \mathbf{X}, \alpha) + \lambda_1 \cdot ||g_\theta(\mathbf{X})||_1 + \lambda_{TV} \cdot ||g_\theta(\mathbf{X})||_{TV} \right]. \tag{8}$$

This method is also popular [18, 19, 50] as it handles structured data well and requires training a single model $g_\theta(\mathbf{x})$ that outputs explanations rather than repeatedly solving Eq. (7) for each sample.

# 7. Experiments

We showcase different aspects of our theoretical findings across multiple settings: a synthetic example, sentiment analysis on the SemEval Twitter corpus [52], and high-dimensional image classification using the CelebA-HQ [53] and RSNA CT scan [54] datasets. The code to reproduce these experiments is available at `https://github.com/Sulam-Group/Sufficient-vs.-Necessary-Explanations`

## 7.1. Synthetic Setting

We consider features $\mathbf{X} \in \mathbb{R}^7$, where $X_i \sim \mathcal{N}(0, 1)$ for $i \in \{1, 4, 5, 6, 7\}$. The remaining $X_i$ and response $Y$ follow, $X_2 = X_1 + \epsilon_1, Y = X_2 + \epsilon_2, X_3 = 5 \cdot Y + 5 \cdot X_4 + \epsilon_3$ for $\epsilon_i \sim \mathcal{N}(0, 1)$. The data-generating process is represented by the directed acyclic graph (DAG) shown in Fig. 1 (note $X_5, X_6$ and $X_7$ are omitted since they share no dependencies with any other $X_i$ or $Y$). In this setting, $Y \perp\!\!\!\perp \mathbf{X}_{\{1,5,6,7\}}|\mathbf{X}_{2,3,4}$ and $Y \perp\!\!\!\perp \mathbf{X}_{\{4,5,6,7\}}$. Thus, for $f(\mathbf{X}) = \mathbb{E}[Y \mid \mathbf{X}]$ and reference $\mathcal{V}_S = p(\mathbf{X}_{S^c} \mid \mathbf{x}_S)$, the solutions to ($P_{suf}$) and ($P_{nec}$) for $\tau = 3$ are $S_{suf}^* = \{2, 3, 4\}$ and $S_{nec}^* = \{1, 2, 3\}$.

---

[1]Here, $\lambda_1, ||S||_1$ and $\lambda_{TV}, ||S||_{TV}$ are the $\ell_1$ and Total Variation norms and hyperparamters, respectively, promoting sparsity and smoothness.

**Validation of Solutions.** For 1000 samples $\mathbf{x}$, we compute solutions to ($P_{\text{suf}}$), ($P_{\text{nec}}$), and ($P_{\text{uni}}$) ($\alpha = 1/2$) for $\tau = 3$ via an exhaustive search. We denote the solutions $\hat{S}_{\text{suf}}$, $\hat{S}_{\text{nec}}$ and $\hat{S}_{\text{uni}}$. For all $\mathbf{x}$, $\hat{S}_{\text{suf}} = S^*_{\text{suf}}$ and $\hat{S}_{\text{nec}} = S^*_{\text{nec}}$, as expected. However, $\hat{S}_{\text{uni}}$ varies. In Fig. 2, we plot the prevalence of the three most reoccurring solutions $\hat{S}_{\text{uni}}$: $\{1, 2, 3\}, \{2, 3, 4\}$, or $\{1, 3, 4\}$. For most $\mathbf{x}$, $S^*_{\text{suf}}$ or $S^*_{\text{nec}}$ are also solutions to ($P_{\text{uni}}$), however for $\approx 7\%$ of samples, $\hat{S}_{\text{uni}} = \{1, 3, 4\}$ is the optimal solution to $P_{\text{uni}}$, which illustrates how solutions to these problems are highly input specific.

**Analysis of Post-hoc Methods.** For all $\mathbf{x}$ with $\hat{S}_{\text{uni}} = \{1, 3, 4\}$, we compute importance scores for each feature using Shapley values (SV), Integrated Gradients (IG) [55], GradientSHAP (GS) [29], and LIME [8]. For each method, we construct sets $\tilde{S}$ by picking the three highest scoring features. In Table 1, we report the $\tilde{S}$ returned by different methods. We see that all methods, except the Shapley value, assign high scores to features in $S^*_{\text{suf}}$. Thus, many methods effectively identify sufficient sets. On the other hand, for approximately 70% of samples, the Shapley value assigns high scores to features in $S^*_{\text{uni}}$. Therefore, the Shapley value often identifies sufficient and necessary features. This suggests that measuring how much a feature contributes to all subsets, as Shapley does, implicitly measures whether a feature is a member of a sufficient and necessary set.

## 7.2. Natural Language Sentiment Classification

We consider a sentiment analysis task on tweets in the SemEval-2017 dataset [52]. The model is a RoBERTa language model [56] that predicts a tweet's sentiment as positive, negative, or neutral. We work in the token space thus our features are text tokens produced by the RoBERTa tokenizer.

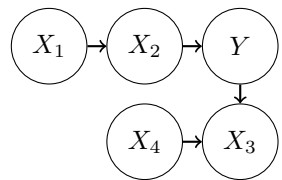

Figure 1: DAG modeling the synthetic data-generating process.

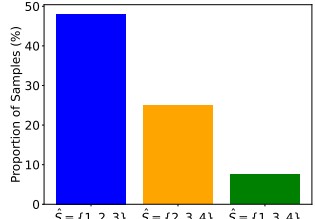

Figure 2: Prevalence of different solutions to $P_{\text{uni}}$ in synthetic setting.

Table 1: Performance of common post-hoc methods on synthetic setting.

|  | Most Prevalent $\tilde{S}$ | % of Samples |
|---|---|---|
| IG | {2,3,4} | 100 |
| GS | {2,3,4} | 100 |
| LIME | {2,3,4} | 100 |
| SV | {1,3,4} | 72 |

**Analysis of Post-hoc Methods.** For a holdout set of tweets classified with either a positive or negative sentiment and containing at most 25 tokens, we solve ($P_{\text{suf}}$), ($P_{\text{nec}}$), and ($P_{\text{uni}}$) via exhaustive search for $\tau = \lceil d\rho \rceil$, where $d$ is the number of tokens in the tweet and $\rho \in \{0.05, 0.10, \ldots, 0.45\}$. Additionally, we use Integrated Gradients (IG) [55] and GradientSHAP (GS) [29] to generate importance scores for each token. To identify if these methods identify features that are sufficient, necessary, or both, we compare how similar the set of features $\hat{S}$, generated by selecting features with the top $\lceil d\rho \rceil$ scores, is to the optimal sets $S^*_{\text{suf}}$, $S^*_{\text{nec}}$, and $S^*_{\text{uni}}$. In Table 2, for $\rho \in \{0.20, 0.25, 0.30\}$, we report the Jaccard Index [57], $J$, between the sets generated by Integrated Gradients and GradientShap and the optimal sets. Here, we see that both Integrated Gradients and GradientShap rank features based on their sufficiency *and* necessity, as indicated by the Jaccard Index between $\hat{S}$ and $S^*_{\text{uni}}$ being the highest across different values of $\rho$.

Table 2: Jaccard Index between the sets generated by Integrated Gradients and GradientShap and the optimal solutions $S^*_{\text{suf}}$, $S^*_{\text{nec}}$, and $S^*_{\text{uni}}$ for tweets from the SemEval-2017 dataset.

|  | $\rho = 0.20$ | | $\rho = 0.25$ | | $\rho = 0.30$ | |
|---|---|---|---|---|---|---|
|  | IG | GS | IG | GS | IG | GS |
| $J(\hat{S}, S^*_{\text{suf}})$ | $0.65 \pm 0.06$ | $0.58 \pm 0.05$ | $0.63 \pm 0.05$ | $0.58 \pm 0.05$ | $0.57 \pm 0.03$ | $0.55 \pm 0.04$ |
| $J(\hat{S}, S^*_{\text{nec}})$ | $0.64 \pm 0.06$ | $0.57 \pm 0.06$ | $0.59 \pm 0.06$ | $0.54 \pm 0.05$ | $0.53 \pm 0.05$ | $0.51 \pm 0.05$ |
| $J(\hat{S}, S^*_{\text{uni}})$ | $\mathbf{0.69 \pm 0.05}$ | $\mathbf{0.62 \pm 0.06}$ | $\mathbf{0.64 \pm 0.05}$ | $\mathbf{0.59 \pm 0.06}$ | $\mathbf{0.60 \pm 0.04}$ | $\mathbf{0.57 \pm 0.04}$ |

Table 3: Comparison of solutions $S^*_{\text{suf}}$, $S^*_{\text{nec}}$, and $S^*_{\text{uni}}$ on the SemEval-2017 dataset.

| | $\rho = 0.20$ | $\rho = 0.25$ | $\rho = 0.30$ | $\rho = 0.35$ |
|---|---|---|---|---|
| $J(S^*_{\text{suf}}, S^*_{\text{nec}})$ | $0.55 \pm 0.05$ | $0.54 \pm 0.05$ | $0.54 \pm 0.05$ | $0.55 \pm 0.05$ |
| $J(S^*_{\text{suf}}, S^*_{\text{uni}})$ | $0.71 \pm 0.06$ | $0.68 \pm 0.06$ | $0.66 \pm 0.05$ | $0.64 \pm 0.05$ |
| $J(S^*_{\text{nec}}, S^*_{\text{uni}})$ | $0.73 \pm 0.07$ | $0.71 \pm 0.07$ | $0.67 \pm 0.07$ | $0.65 \pm 0.07$ |

**Sufficient Solution: $S^*_{\text{suf}}$**

Time warner `is` the devil. `Worst` possible time for the Internet to go out .

**Necessary Solution: $S^*_{\text{nec}}$**

Time warner is the `devil` . `Worst` possible time for the Internet to go out.

**Unified Solution: $S^*_{\text{uni}}$**

Time warner is the `devil` . `Worst` possible time for the Internet to go out.

Figure 3: Solutions ($\rho = 0.10$), $S^*_{\text{suf}}$, $S^*_{\text{nec}}$, and $S^*_{\text{uni}}$, for a tweet from the SemEval-2017 dataset.

**Sufficiency vs Necessity**. We also quantiy the difference between $S^*_{\text{suf}}$, $S^*_{\text{nec}}$, and $S^*_{\text{uni}}$. In Table 3, we report the Jaccard Index between these sets for various values of $\rho$. Observe that for all $\rho$, $S^*_{\text{suf}}$ and $S^*_{\text{nec}}$ exhibit the lowest Jaccard Index, indicating that these sets are highly dissimilar. On the other hand, as expected, the Jaccard Index between $S^*_{\text{uni}}$ and $S^*_{\text{suf}}$ or $S^*_{\text{nec}}$ is much higher, as the solutions $S^*_{\text{uni}}$, by construction, balance sufficiency and necessity. In Fig. 3, we present example solutions for a tweet classified with a negative sentiment to highlight the differences. The solutions differ: $S^*_{\text{suf}}$ consists of the words `Worst` and `is` as sufficient. However, $S^*_{\text{nec}}$ and $S^*_{\text{uni}}$ both contain `Worst` and `devil`, as removing both words is necessary for the tweet to lose its negative sentiment. This example illustrates how sufficient and necessary sets can differ while providing equally valuable insights into how models make predictions. Additional results and examples are in Appendix A.4.

### 7.3. Image Classification

We consider two image classification tasks on the CelebA-HQ [58] and RSNA 2019 Brain CT Hemorrhage Challenge [54] datasets. The RSNA results are deferred to Appendix A.2. In both experiments the features are pixel values and so a subset $S$ corresponds to a binary mask that identifies a set pixels. With these experiments, we will analyze the ability of popular explanation methods–including Integrated Gradients [55], GradientSHAP [29], Guided GradCAM [8], and h-Shap [13]–to identify small sufficient and necessary subsets. To ensure consistent analysis, all attribution scores are normalized to the interval $[0, 1]$. This is done by setting the top 1% of nonzero scores to 1 and dividing the remaining by the minimum score from the top 1% nonzero scores, which is common practice [59]. Binary masks are then generated by thresholding the normalized scores using thresholds $t \in (0, 1)$. For a test set of images and normalized attribution scores, we report the average (across all binary masks) $-\log(\Delta^{\text{suf}})$, $-\log(\Delta^{\text{nec}})$, and $-\log(L^0)$ where $L^0$ is the relative size of $S$ for $t \in (0, 1)$ to analyze the sufficiency, necessity and size of the explanations. Additionally, we will demonstrate and visualize the similarities and differences between sufficient and necessary sets.

#### 7.3.1. CelebA-HQ

We use a modified version of the CelebA-HQ dataset with 30,000 celebrity faces resized to $256 \times 256$. The model is a ResNet18 that predicts whether a celebrity is smiling with $\approx 94\%$ test accuracy. To generate sufficient and necessary masks, we use the model based approach and learn sufficient and necessary explainer models. Given the structured nature of the data and the similarity of features across images, we use this approach because it prevents overfitting to spurious signals [50], an issue that can arise with per-example methods. Implementation details are included in Appendix A.3.

**Analysis of Post-hoc Methods.** For 100 images correctly classified by the ResNet model, we apply multiple post-hoc methods and our explainers to identify important features associated with smil-

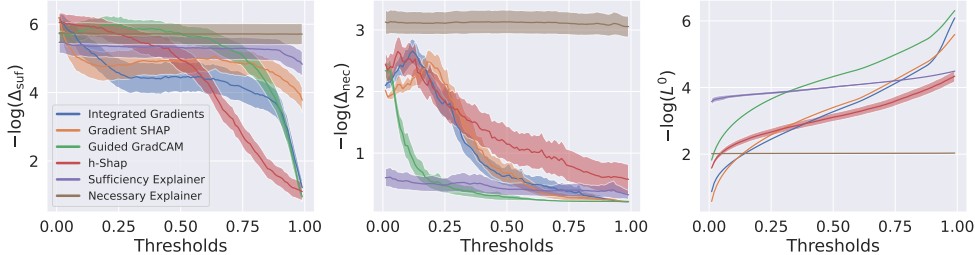

Figure 4: Comparison of different explanation methods on the CelebA-HQ dataset.

ing. Fig. 4 illustrates that for a wide range of thresholds $t \in [0, 1]$, many methods identify sufficient subsets, as $-\log(\Delta^{\mathsf{suf}})$ for many of them is comparable to that of the sufficient explainer. The necessary explainer, in fact, identifies subsets that are more sufficient than those found by the sufficient explainer. The reason is that the sufficient explainer identifies subsets that are, on average, smaller for all $t \in [0, 1]$, while the necessary explainer finds subsets that are constant in size for all $t$ but slightly larger since, to be necessary, they must contain more features that provide additional information. For other methods, as $t$ increases, subset size decreases, and the sufficiency and necessity of the solutions decline. Meanwhile, the necessary explainer naturally identifies necessary subsets, indicated by large $-\log(\Delta^{\mathsf{nec}})$, whereas other methods fail to do so. In conclusion, many methods can identify sufficient sets, but not necessary ones and directly optimizing for these criterion leads to identifying small, constant-sized subsets across thresholds.

**Sufficiency vs. Necessity.** In Fig. 5, we observe that sufficient subsets alone may miss important features, whereas solutions to $(P_{\mathsf{uni}})$ offer deeper insights. As stated earlier, the sufficient explainer identifies sets that are sufficient but not necessary. On the other hand, the necessary explainer exhibits high $-\log(\Delta^{\mathsf{suf}})$ and $-\log(\Delta^{\mathsf{nec}})$, indicating that it identifies both sufficient and necessary sets, i.e. solutions to $(P_{\mathsf{uni}})$. In Fig. 5, we visualize the reasons for this phenomenon. Notice that $S^*_{\mathsf{suf}}$ precisely highlights (only) the smile. When $S^*_{\mathsf{suf}}$ is kept, one can generate new images (as done in [46]) on which the model also predicts smile. On the other hand, we see why $S^*_{\mathsf{suf}}$ is *not* necessary: by keeping its complement, $(S^*_{\mathsf{suf}})_c$, we preserve important features that lead to new images with smiles, leading the model to produce the same prediction as it did for the original image. Conversely solutions to $(P_{\mathsf{nec}})$ (also solutions to $(P_{\mathsf{uni}})$ here) generate different explanations that provide a more complete picture of feature importance. Notice that $S^*_{\mathsf{nec}}$ is sufficient because $S^*_{\mathsf{suf}} \subseteq S^*_{\mathsf{nec}}$, with the additional features mainly being the dimples and eyes, which aid in determining the presence of a smile. More importantly, Fig. 6 illustrates why $S^*_{\mathsf{nec}}$ is necessary: when we fix the complement of $S^*_{\mathsf{nec}}$ and generate new samples, the face may lack a smile, leading the model to predict no smile. Additional images and details on sample generation are in Appendices A.3 and A.4.

## 8. Limitations & Broader Impacts

While this work provides a novel theoretical contribution to the XAI community, there are some limitations that require careful discussion. The choice of reference distribution $\mathcal{V}_S$ is crucial. For example, only with the conditional distribution can one obtain the independence results that our theory provides. Naturally, there are computational trade-offs that must be studied; the ability to learn and sample from accurate conditional distributions to generate explanations with clear statistical meaning comes with a computational and statistical cost, particularly in high-dimensional settings. Thus, a direction for future work is to explore the impact of different $\mathcal{V}_S$ and provide a principled framework for selecting one that balances practical utility and computational feasibility.

Another relevant question is how well our proposed notions align with human intuition. While we aim to understand which features are sufficient and necessary *for a given model*, these explanations may not always align with how humans perceive importance. This can be an issue in settings where interpretability is essential for trust and accountability. On the one hand, our approach provides useful insights to further evaluate models (e.g. by verifying if the sufficient and necessary features correlate with the correct ones as informed by human experts). On the other hand, bridging the gap

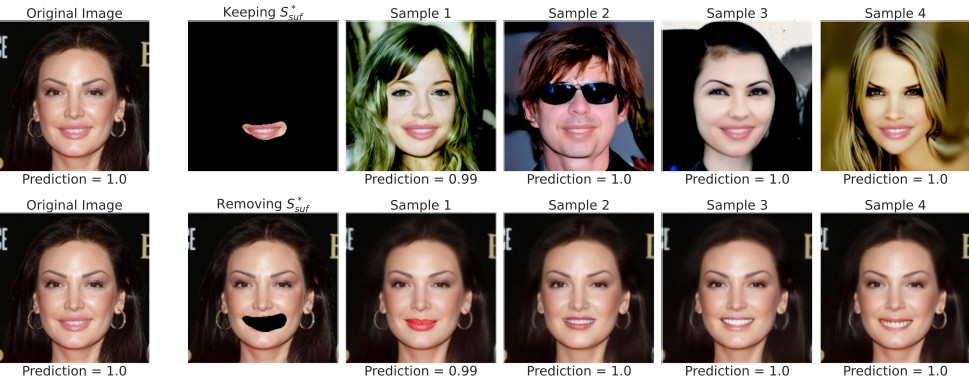

Figure 5: Images and model predictions by keeping and removing the sufficient subset $S_{\mathsf{suf}}^*$.

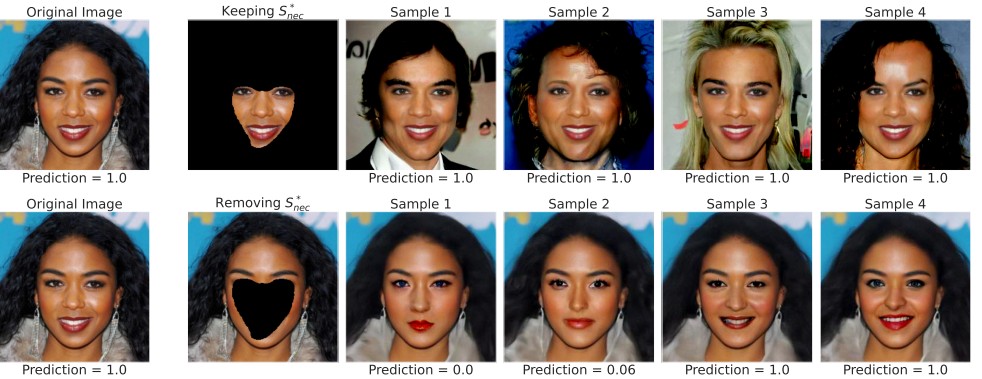

Figure 6: Images and model predictions by keeping and removing the necessary subset $S_{\mathsf{nec}}^*$.

between our definitions and other human notions of importance is an area for further investigation. User studies and collaboration with domain experts will be critical in determining how our formal notions can be adapted to better meet real-world interpretability needs. Finally, the societal impact of this work warrants discussion. While we offer a rigorous framework to understand model predictions, these are oblivious to notions of demographic bias [60–62]. There is a risk that an "incorrect" choice of sufficient vs. necessary explanation could reinforce biases or obscure the causal reasons behind predictions. Future work will study how our framework can incorporate these biases.

## 9. Conclusion

This work formalizes notions of sufficiency and necessity as tools to evaluate feature importance and explain model predictions. We demonstrate that sufficient and necessary explanations, while insightful, often provide incomplete while complementary answers to model behavior. To address this limitation, we propose a unified approach that offers a new and more nuanced understanding of model behavior. Our unified approach expands the scope of explanations and reveals trade-offs between sufficiency and necessity, giving rise to new interpretations of feature importance. Through our theoretical contributions, we present conditions under which sufficiency and necessity align or diverge, and provide two perspectives of our unified approach through the lens of conditional independence and Shapley values. Our experimental results support our theoretical findings, providing examples of how adjusting sufficiency-necessity trade-off via our unified approach can uncover alternative sets of important features that would be missed by focusing solely on sufficiency or necessity. Furthermore, we evaluate common post-hoc interpretability methods showing that many fail to reliably identify features that are necessary or sufficient. In summary, our work contributes to a more complete understanding of feature importance through sufficiency and necessity. We believe, and hope, our framework holds potential for advancing the rigorous interpretability of ML models.

## Acknowledgements

This research was supported in part by NSF CAREER Award CCF 2239787 and NIH award R01CA287422.

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

# A. Appendix

## A.1. Proofs

### A.1.1. Proof of Lemma 4.1

**Lemma 4.1.** Let $\alpha \in (0, 1)$. For $\tau > 0$, denote $S^*$ to be a solution to $(P_{uni})$ for which $\Delta_{\mathcal{V}}^{uni}(S^*, f, \mathbf{x}, \alpha) = \epsilon$. Then, $S^*$ is $\frac{\epsilon}{\alpha}$-sufficient and $\frac{\epsilon}{1-\alpha}$-necessary. Formally,

$$0 \leq \Delta_{\mathcal{V}}^{suf}(S^*, f, \mathbf{x}) \leq \frac{\epsilon}{\alpha} \quad \text{and} \quad 0 \leq \Delta_{\mathcal{V}}^{nec}(S^*, f, \mathbf{x}) \leq \frac{\epsilon}{1-\alpha}. \tag{9}$$

*Proof.* Let $\tau > 0$ and $\alpha \in (0, 1)$ and denote $S^*$ to be a solution to $(P_{uni})$ such that

$$\Delta_{\mathcal{V}}^{uni}(S^*, f, \mathbf{x}, \alpha) = \epsilon. \tag{10}$$

Then, by definition of being a solution to $(P_{uni})$,

$$|S^*| \leq \tau. \tag{11}$$

Furthermore, recall that

$$\Delta_{\mathcal{V}}^{uni}(S^*, f, \mathbf{x}, \alpha) = \alpha \cdot \Delta_{\mathcal{V}}^{suf}(S^*, f, \mathbf{x}) + (1 - \alpha) \cdot \Delta_{\mathcal{V}}^{nec}(S^*, f, \mathbf{x}) \tag{12}$$

which implies

$$\alpha \cdot \Delta_{\mathcal{V}}^{suf}(S^*, f, \mathbf{x}) = \epsilon - (1 - \alpha) \cdot \Delta_{\mathcal{V}}^{nec}(S^*, f, \mathbf{x}) \tag{13}$$

$$\leq \epsilon \qquad\qquad ((1 - \alpha), \Delta_{\mathcal{V}}^{nec}(S^*, f, \mathbf{x}) \geq 0) \tag{14}$$

$$\implies \Delta_{\mathcal{V}}^{suf}(S^*, f, \mathbf{x}) \leq \frac{\epsilon}{\alpha}. \tag{15}$$

Similarly,

$$(1 - \alpha) \cdot \Delta_{\mathcal{V}}^{nec}(S^*, f, \mathbf{x}) = \epsilon - \alpha \cdot \Delta_{\mathcal{V}}^{suf}(S^*, f, \mathbf{x}) \tag{16}$$

$$\leq \epsilon \qquad\qquad (\alpha, \Delta_{\mathcal{V}}^{suf}(S^*, f, \mathbf{x}) \geq 0) \tag{17}$$

$$\implies \Delta_{\mathcal{V}}^{nec}(S^*, f, \mathbf{x}) \leq \frac{\epsilon}{1-\alpha}. \tag{18}$$

$\square$

### A.1.2. Proof of Lemma 4.2

**Lemma 4.2.** For $0 \leq \epsilon < \frac{\rho(f(\mathbf{x}), f_\emptyset(\mathbf{x}))}{2}$, denote $S_{suf}^*$ and $S_{nec}^*$ to be $\epsilon$-sufficient and $\epsilon$-necessary sets. Then, if $S_{suf}^*$ is $\epsilon$-super sufficient or $S_{nec}^*$ is $\epsilon$-super necessary,

$$S_{suf}^* \cap S_{nec}^* \neq \emptyset. \tag{19}$$

*Proof.* We will prove the result via contradiction. First recall that,

$$f_S(\mathbf{x}) = \mathop{\mathbb{E}}_{\mathbf{X}_{S^c} \sim \mathcal{V}_{S^c}} [f(\mathbf{x}_S, \mathbf{X}_{S^c})] \tag{20}$$

and, for any metric $\rho : \mathbb{R} \times \mathbb{R} \mapsto \mathbb{R}$,

$$\Delta_{\mathcal{V}}^{suf}(S, f, \mathbf{x}) \triangleq \rho(f(\mathbf{x}), f_S(\mathbf{x})) \tag{21}$$

$$\Delta_{\mathcal{V}}^{nec}(S, f, \mathbf{x}) \triangleq \rho(f_{S^c}(\mathbf{x}), f_\emptyset(\mathbf{x})). \tag{22}$$

Since $\rho$ is a metric on $\mathbb{R}$, it satisfies the triangle inequality. Thus, for $a, b, c \in \mathbb{R}$

$$\rho(a, c) \leq \rho(a, b) + \rho(b, c). \tag{23}$$

Now, let $S_{suf}^*$ be $\epsilon$-super sufficient and suppose

$$S_{suf}^* \cap S_{nec}^* = \emptyset. \tag{24}$$

This implies

$$S_{\text{suf}}^* \subseteq (S_{\text{nec}}^*)_c. \tag{25}$$

Subsequently, since $S_{\text{suf}}^*$ is $\epsilon$-super sufficient,

$$\Delta_{\mathcal{V}}^{\text{suf}}((S_{\text{nec}}^*)_c, f, \mathbf{x}) \leq \epsilon. \tag{26}$$

As a result, observe

$$
\begin{aligned}
\rho(f(\mathbf{x}), f_{\emptyset}(\mathbf{x})) &\leq \rho(f(\mathbf{x}), f_{(S_{\text{nec}}^*)_c}(\mathbf{x})) + \rho(f_{(S_{\text{nec}}^*)_c}(\mathbf{x}), f_{\emptyset}(\mathbf{x})) && \text{triangle inequality} && (27) \\
&= \Delta_{\mathcal{V}}^{\text{suf}}((S_{\text{nec}}^*)_c, f, \mathbf{x}) + \Delta_{\mathcal{V}}^{\text{nec}}((S_{\text{nec}}^*), f, \mathbf{x}) && && (28) \\
&\leq \epsilon + \Delta_{\mathcal{V}}^{\text{nec}}((S_{\text{nec}}^*), f, \mathbf{x}) && S_{\text{suf}}^* \text{ is } \epsilon\text{-super sufficient} && (29) \\
&\leq 2\epsilon && S_{\text{nec}}^* \text{ is } \epsilon\text{-necessary} && (30) \\
\implies \epsilon &\geq \frac{\rho(f(\mathbf{x}), f_{\emptyset}(\mathbf{x}))}{2} && && (31)
\end{aligned}
$$

which is a contradiction because $0 \leq \epsilon < \frac{\rho(f(\mathbf{x}), f_{\emptyset}(\mathbf{x}))}{2}$. Thus $S_{\text{suf}}^* \cap S_{\text{nec}}^* \neq \emptyset$. The proof of this result assuming $S_{\text{nec}}^*$ is $\epsilon$-super necessary follows the same argument. $\square$

### A.1.3. Proof of Theorem 4.1

**Theorem 4.1.** Let $\tau_1, \tau_2 > 0$ and $0 \leq \epsilon < \frac{1}{2} \cdot \rho(f(\mathbf{x}), f_{\emptyset}(\mathbf{x}))$. Denote $S_{\text{suf}}^*$ and $S_{\text{nec}}^*$ to be $\epsilon$-super sufficient and $\epsilon$-super necessary solutions to (P$_{\text{suf}}$) and (P$_{\text{nec}}$), respectively, such that $|S_{\text{suf}}^*| = \tau_1$ and $|S_{\text{nec}}^*| = \tau_2$. Then, there exists a set $S^*$ such that

$$\Delta_{\mathcal{V}}^{\text{uni}}(S^*, f, \mathbf{x}, \alpha) \leq \epsilon \quad \text{and} \quad \max(\tau_1, \tau_2) \leq |S^*| < \tau_1 + \tau_2. \tag{32}$$

Furthermore, if $S_{\text{suf}}^* \subseteq S_{\text{nec}}^*$ or $S_{\text{nec}}^* \subseteq S_{\text{suf}}^*$. then $S^* = S_{\text{nec}}^*$ or $S^* = S_{\text{suf}}^*$, respectively.

*Proof.* Consider the set $S^* = S_{\text{suf}}^* \cup S_{\text{nec}}^*$. This set has the following properties:

(P1) $S^*$ is $\epsilon$-sufficient because $S_{\text{suf}}^*$ is $\epsilon$-super sufficient

(P2) $S^*$ is $\epsilon$-necessary because $S_{\text{suf}}^*$ is $\epsilon$-super necessary

(P3) $|S^*| \geq \max(\tau_1, \tau_2)$ with $|S^*| = \tau_1$ when $S_{\text{nec}}^* \subset S_{\text{suf}}^*$ and with $|S^*| = \tau_2$ when $S_{\text{suf}}^* \subset S_{\text{nec}}^*$

(P4) Via Lemma 4.1, we know $S_{\text{suf}}^* \cap S_{\text{nec}}^* \neq \emptyset$ thus $|S^*| < \tau_1 + \tau_2$

Then by (P1) and (P2)

$$
\begin{aligned}
\Delta_{\mathcal{V}}^{\text{uni}}(S^*, f, \mathbf{x}, \alpha) &= \alpha \cdot \Delta_{\mathcal{V}}^{\text{suf}}(S^*, f, \mathbf{x}) + (1 - \alpha) \cdot \Delta_{\mathcal{V}}^{\text{nec}}(S^*, f, \mathbf{x}) && (33) \\
&\leq \alpha \cdot \epsilon + (1 - \alpha) \cdot \epsilon = \epsilon && (34)
\end{aligned}
$$

and by (P3) and (P4) we have $\max(\tau_1, \tau_2) \leq |S^*| < \tau_1 + \tau_2$, $\square$

### A.1.4. Proof of Corollary 5.1

**Corollary 5.1.** Suppose for any $S \subseteq [d]$, $\mathcal{V}_S = p(\mathbf{X}_S \mid \mathbf{X}_{S^c} = \mathbf{x}_{S^c})$. Let $\alpha \in (0, 1)$, $\epsilon \geq 0$, and denote $\rho : \mathbb{R} \times \mathbb{R} \mapsto \mathbb{R}$ to be a metric on $\mathbb{R}$. Furthermore, for $f(\mathbf{X}) = \mathbb{E}[Y \mid \mathbf{X}]$ and $\tau > 0$, let $S^*$ be a solution to (P$_{\text{uni}}$) such that $\Delta_{\mathcal{V}}^{\text{uni}}(S, f, \mathbf{x}, \alpha) = \epsilon$. Then, $S^*$ satisfies the following conditional independence relations,

$$\rho\left(\mathbb{E}[Y \mid \mathbf{x}], \mathbb{E}[Y \mid \mathbf{X}_{S^*} = \mathbf{x}_{S^*}]\right) \leq \frac{\epsilon}{\alpha} \quad \text{and} \quad \rho\left(\mathbb{E}[Y \mid \mathbf{X}_{S_c^*} = \mathbf{x}_{S_c^*}], \mathbb{E}[Y]\right) \leq \frac{\epsilon}{1 - \alpha}. \tag{35}$$

*Proof.* All we need to show is that when $\mathcal{V}_S = p(\mathbf{X}_S \mid \mathbf{X}_{S^c} = \mathbf{x}_{S^c})$ and $f(\mathbf{X}) = \mathbb{E}[Y \mid \mathbf{X}]$, we have

$$f_S(\mathbf{x}) = \mathbb{E}[Y \mid \mathbf{X}_S = \mathbf{x}_S]. \tag{36}$$

Once this is proven, we can simply apply Lemma 4.1.

To this end, we have by assumption that $f(\mathbf{x}) = \mathbb{E}[Y \mid \mathbf{X} = \mathbf{x}]$ and, for any $S \subseteq [d]$, $\mathcal{V}_S = p(\mathbf{X}_S \mid \mathbf{X}_{S^c} = \mathbf{x}_{S^c})$. Then by definition

$$f_S(\mathbf{x}) = \mathbb{E}_{\mathcal{V}_{S^c}}[f(\mathbf{x}_S, \mathbf{X}_{S^c})] = \int_{\mathcal{X}} f(\mathbf{x}_S, \mathbf{X}_{S^c}) \cdot p(\mathbf{X}_{S^c} \mid \mathbf{X}_S = \mathbf{x}_S) \, d\mathbf{X}_{S^c} \tag{37}$$

$$= \int_{\mathcal{X}} \mathbb{E}[Y \mid \mathbf{X}_S = \mathbf{x}_S, \mathbf{X}_{S^c}] \cdot p(\mathbf{X}_{S^c} \mid \mathbf{X}_S = \mathbf{x}_S) \, d\mathbf{X}_{S^c} \tag{38}$$

$$= \int_{\mathcal{X}} \left( \int_{\mathcal{Y}} y \cdot p(y \mid \mathbf{X}_S = \mathbf{x}_S, \mathbf{X}_{S^c}) \, dy \right) \cdot p(\mathbf{X}_{S^c} \mid \mathbf{X}_S = \mathbf{x}_S) \, d\mathbf{X}_{S^c} \tag{39}$$

$$= \int_{\mathcal{Y}} y \left( \int_{\mathcal{X}} p(y, \mathbf{X}_{S^c} \mid \mathbf{X}_S = \mathbf{x}_S) \, d\mathbf{X}_{S^c} \right) dy \tag{40}$$

$$= \int_{\mathcal{Y}} y \cdot p(y \mid \mathbf{X}_S = \mathbf{x}_S) \, dy \tag{41}$$

$$= \mathbb{E}[Y \mid \mathbf{X}_S = \mathbf{x}_S]. \tag{42}$$

By applying Lemma 4.1, we have the desired result. $\qquad\square$

### A.1.5. Proof of Theorem 5.1

**Theorem 5.1.** Consider an input $\mathbf{x}$ for which $f(\mathbf{x}) \neq f_\emptyset(\mathbf{x})$. Denote by $\Lambda_d = \{S, S^c\}$ the partition of $[d] = \{1, 2, \ldots, d\}$, and define the characteristic function to be $v(S) = -\rho(f(\mathbf{x}), f_S(\mathbf{x}))$. Then,

$$\phi_S^{\mathsf{shap}}(\Lambda_d, v) \geq \rho(f(\mathbf{x}), f_\emptyset(\mathbf{x})) - \Delta_{\mathcal{V}}^{\mathsf{uni}}(S, f, \mathbf{x}, \alpha). \tag{43}$$

*Proof.* Before we prove the result, recall the following properties of a metric $\rho$ in the reals:

(P1) $\forall a, b \in \mathbb{R}, \ \rho(a, b) = 0 \iff a = b$

(P2) for $a, b, c \in \mathbb{R}, \quad \rho(a, c) \leq \rho(a, b) + \rho(b, c)$.

Now, for the partition $\Lambda_d = \{S, S^c\}$ of $[d] = \{1, 2, \ldots, d\}$ and characteristic function $v(S) = -\rho(f(\mathbf{x}), f_S(\mathbf{x}))$, $\phi_S^{\mathsf{shap}}(\Lambda_d, v)$ is defined as

$$\phi_S^{\mathsf{shap}}(\Lambda_d, v) = \frac{1}{2} \cdot [v(S \cup S^c) - v(S^c)] + \frac{1}{2} \cdot [v(S) - v(\emptyset)] \tag{44}$$

$$= \frac{1}{2} \cdot [\rho(f(\mathbf{x}), f_{S^c}(\mathbf{x})) - \rho(f(\mathbf{x}), f(\mathbf{x}))] + \frac{1}{2} \cdot [\rho(f(\mathbf{x}), f_\emptyset(\mathbf{x})) - \rho(f(\mathbf{x}), f_S(\mathbf{x}))] \tag{45}$$

$$= \frac{1}{2} \cdot [\rho(f(\mathbf{x}), f_{S^c}(\mathbf{x}))] + \frac{1}{2} \cdot [\rho(f(\mathbf{x}), f_\emptyset(\mathbf{x})) - \rho(f(\mathbf{x}), f_S(\mathbf{x}))] \qquad \text{by (P1)} \tag{46}$$

By (P2)

$$\rho(f(\mathbf{x}), f_\emptyset(\mathbf{x})) \leq \rho(f(\mathbf{x}), f_{S^c}(\mathbf{x})) + \rho(f_{S^c}(\mathbf{x}), f_\emptyset(\mathbf{x})) \tag{47}$$

$$\implies \rho(f(\mathbf{x}), f_{S^c}(\mathbf{x})) \geq \rho(f(\mathbf{x}), f_\emptyset(\mathbf{x})) - \rho(f_{S^c}(\mathbf{x}), f_\emptyset(\mathbf{x})). \tag{48}$$

Thus

$$\phi_S^{\mathsf{shap}}(\Lambda_d, v) = \frac{1}{2} \cdot [\rho(f(\mathbf{x}), f_{S^c}(\mathbf{x}))] + \frac{1}{2} \cdot [\rho(f(\mathbf{x}), f_\emptyset(\mathbf{x})) - \rho(f(\mathbf{x}), f_S(\mathbf{x}))] \tag{49}$$

$$\geq \frac{1}{2} \cdot [\rho(f(\mathbf{x}), f_\emptyset(\mathbf{x})) - \rho(f_{S^c}(\mathbf{x}), f_\emptyset(\mathbf{x}))] + \frac{1}{2} \cdot [\rho(f(\mathbf{x}), f_\emptyset(\mathbf{x})) - \rho(f(\mathbf{x}), f_S(\mathbf{x}))] \tag{50}$$

$$= \rho(f(\mathbf{x}), f_\emptyset(\mathbf{x})) - \Delta_{\mathcal{V}}^{\mathsf{uni}}(S, f, \mathbf{x}, \alpha). \tag{51}$$

$\square$

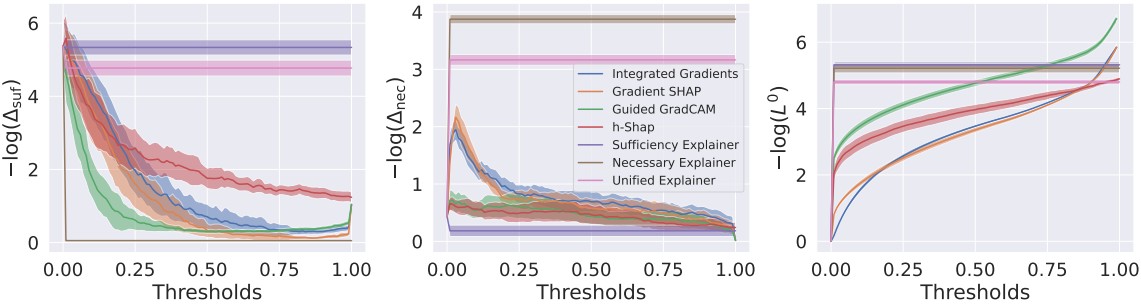

Figure 7: Comparison of different methods on RSNA dataset

## A.2. Additional Experiments

### A.2.1. RSNA CT Hemorrhage

We use the RSNA 2019 Brain CT Hemorrhage Challenge dataset comprised of 752,803 scans. Each scan is annotated by expert neuroradiologists with the presence and type(s) of hemorrhage (i.e., epidural, intraparenchymal, intraventricular, subarachnoid, or subdural). We use a ResNet18 [63] classifier that was pretrained on this data [13]. Since the dataset consists of highly complex and diverse images, we employ the per-example approach in Eq. (7) with $\alpha \in \{0, 0.5, 1\}$ to learn sufficient and necessary masks. Further details are in Appendix A.3.

**Comparison of Post-hoc Interpretability Methods.** For a set of 20 images positively classified by the ResNet model, we apply multiple post-hoc interpretability methods, as well as compute sufficient and necessary masks by solving (7). The results in Fig. 7 show that for thresholds $t < 0.1$, many methods identify sufficient sets smaller in size than the sufficient and unified explainer, as indicated by their large values of $-\log(\Delta^{\mathsf{suf}})$ and smaller values of $-\log(L^0)$. However, for $t > 0.1$, only the sufficient and unified explainer identify sufficient sets of a constant small size. Importantly, *no methods, besides the necessity and unified explainers, identify necessary sets*. Furthermore, as expected, the sufficient explainer does not identify necessary sets and vice versa. The unified explainer, as expected, identifies a sufficient and necessary set (at the cost of a larger set). In conclusion, while off-the-shelf methods can identify sufficient, they do not identify necessary sets for small thresholds. Only by optimizing for such properties one gets explanations that are consistently small, sufficient and/or necessary across thresholds.

**Sufficiency vs. Necessity.** In Fig. 8 we visualize the sufficient and necessary features in various CT scans. The first observation is that sufficient subsets do not provide a complete picture of which features are important. Notice for all the CT scans, a sufficient set, $S_{\mathsf{suf}}^*$ highlights one or two, but never all, brain hemorrhages in the scans. For example, in the last row, $S_{\mathsf{suf}}^*$ only contains the right frontal lobe parenchymal hemorrhages, which happens to be one of the larger hemorrhages present. On the other hand, necessary sets, $S_{\mathsf{nec}}^*$, contain parts of, sometimes entirely, *all* hemorrhages in the scans. In the last row, $S_{\mathsf{nec}}^*$ contains all multifocal parenchymal hemorrhages in both right and left frontal lobes, because when all these regions are masked, the model yields a prediction $\approx 0.64$– the prediction of the model on the mean image. Finally, notice in the 2nd and 3rd columns that $S_{\mathsf{nec}}^*$ and $S_{\mathsf{uni}}^*$ are nearly identical, which precisely demonstrate Lemma 4.1 and Theorem 4.1 in practice. First, since $S_{\mathsf{suf}}^*$ is super sufficient, $S_{\mathsf{suf}}^*$ and $S_{\mathsf{nec}}^*$, share common features. Second, visually $S_{\mathsf{suf}}^* \subseteq S_{\mathsf{nec}}^*$ holds approximately and so $S_{\mathsf{nec}}^* = S_{\mathsf{uni}}^*$. Through this experiment we are able to highlight the differences between sufficient and necessary sets, show how each contain important and complementary information, and demonstrate our theory holding in real world settings.

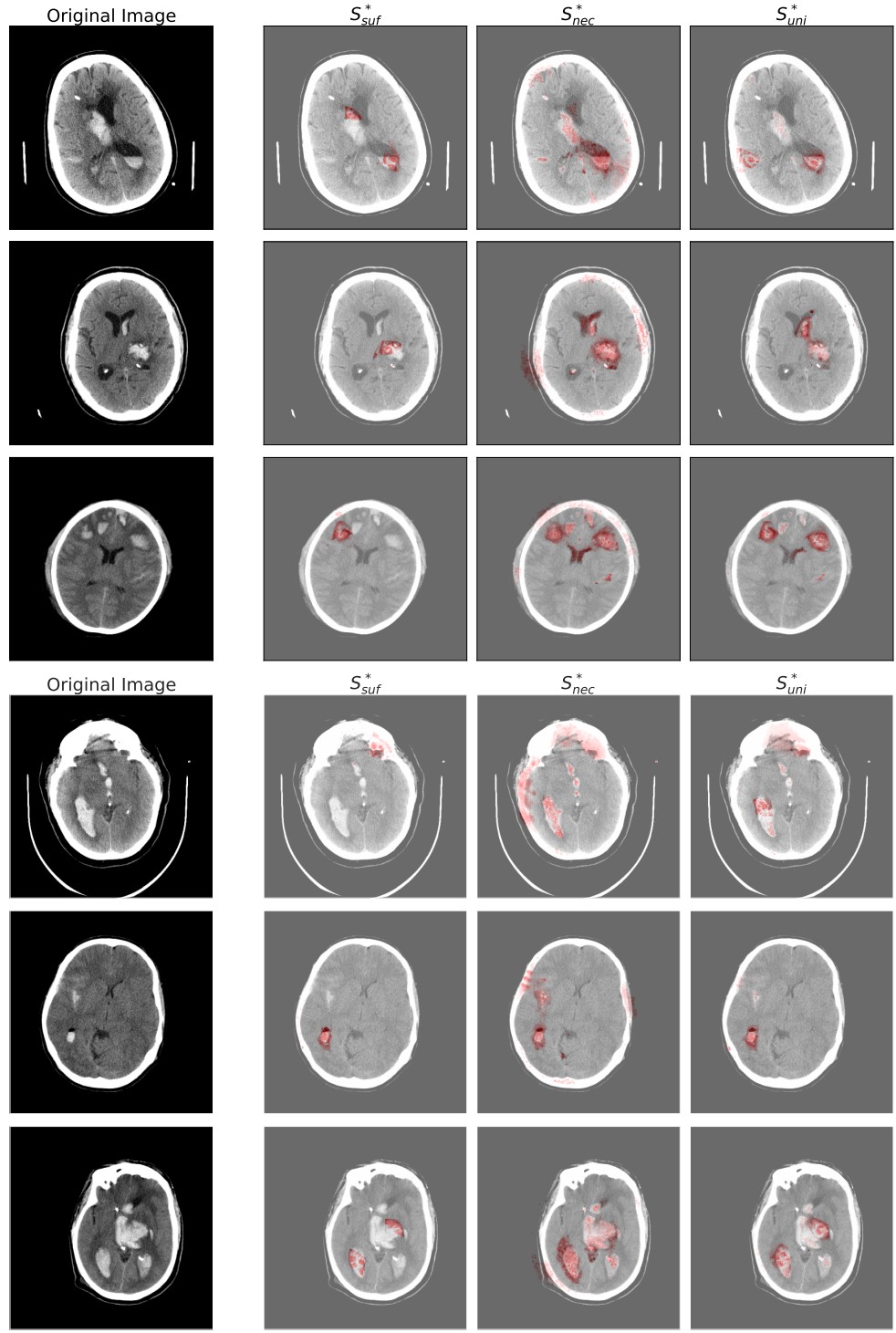

Figure 8: $S^*_{\mathsf{suf}}$, $S^*_{\mathsf{nec}}$ and $S^*_{\mathsf{uni}}$ for various CT scans.

### A.3. Additional Experimental Details

In this section, we include further experimental details. All experiments were performed on a private cluster with 8 NVIDIA RTX A5000 with 24 GB of memory. All scripts were run on PyTorch `2.0.1`, Python `3.11.5`, and CUDA `12.2`.

### A.3.1. RSNA CT Hemorrhage

**Dataset Details.** The RSNA 2019 Brain CT Hemorrhage Challenge dataset [54], contains 75,2803 images labeled by a panel of board-certified radiologists with the types of hemorrhage present (epidural, intraparenchymal, intraventricular, subarachnoid, subdural).

**Implementation.** For this experiment we solve the relaxed optimization problem [11, 16]

$$\arg\min_{S \subseteq [0,1]^d} \Delta_{\mathcal{V}}^{\mathsf{uni}}(S, f, \mathbf{x}, \alpha) + \lambda_1 \cdot ||S||_1 + \lambda_{\mathrm{TV}} \cdot ||S||_{TV}. \tag{52}$$

where

$$\Delta_{\mathcal{V}}^{\mathsf{uni}}(g_\theta(\mathbf{x}_i), f, \mathbf{x}_i, \alpha) = \alpha \cdot |f(\mathbf{x}_i) - f_S(\mathbf{x}_i)| + (1 - \alpha) \cdot |f(\mathbf{x}_i) - f_S(\mathbf{x}_i)| \tag{53}$$

to identify sufficient and necessary masks $S$ for a sample $\mathbf{x}$. Here $||S||_1$ and $||S||_{TV}$ are the $L^1$ and Total Variation norm of $S$, which promote sparsity and smoothness respectively and $\lambda_{\mathsf{Sp}}$ and $\lambda_{\mathsf{Sm}}$ are the associated. To solve this problem, a mask $S \in [0,1]^{512 \times 512}$ is initialized with entries $S_i \sim \mathcal{N}(0.5, \frac{1}{36})$. For 1000 iterations, the mask $S$ is iteratively updated to minimize the objective function above, where for any $S$,

$$f_S(\mathbf{x}) = \frac{1}{K} \sum_{i=1}^{K} f((\tilde{\mathbf{X}}_S)_i) \quad \text{with} \quad (\tilde{\mathbf{X}}_S)_i = \mathbf{x} \circ \tilde{\mathbb{1}}_S + (1 - \tilde{\mathbb{1}}_S) \circ b_i. \tag{54}$$

Here the entries $(\tilde{\mathbb{1}}_S)_i \sim \mathrm{Bernoulli}(S_i)$ and $b_i$ is the $i$th entry of a vector $\mathbf{b} = (b_1, \cdots, b_d) \sim \mathcal{V}$. For this experiment, the reference distribution $\mathcal{V}$ is the unconditional mean image over the set of training images. Therefore $b_i$ is the average value of the $i$th pixel over the training set. To allow for differentiation during optimization, we generate discrete samples $\tilde{\mathbb{1}}_S$ using the Gumbel-Softmax distribution. With this formulation, the entries $(\tilde{\mathbf{X}}_S)_i$ follow a Bernoulli distribution with outcomes $\{b_i, x_i\}$, i.e. $(\tilde{\mathbf{X}}_S)_i$ is distributed as

$$\Pr[(\tilde{\mathbf{X}}_S)_i = x_i] = S_i \quad \text{and} \quad \Pr[(\tilde{\mathbf{X}}_S)_i = b_i] = 1 - S_i. \tag{55}$$

For every $\alpha \in \{0, 0.5, 1\}$, during optimization we set $K = 10$, $\lambda_1 = 3$ and $\lambda_{\mathrm{TV}} = 20$. We utilize the Adam optimizer with default $\beta$-parameters of $\beta_1 = 0.9$, $\beta_2 = 0.99$ and a fixed learning rate of 0.01.

### A.3.2. CelebA-HQ

**Dataset Details.** We use a modified version of the CelebA-HQ dataset [53, 58] which contains 30,000 celebrity faces resized to 256×256 pixels with several landmark locations and binary attributes (e.g., eyeglasses, bangs, smiling).

**Implementation.** Recall for this experiment, to generate sufficient or necessary masks $S$ for samples $\mathbf{x}$, we learn a model $g_\theta : \mathcal{X} \mapsto [0,1]^d$ via solving the following optimization problem:

$$\arg\min_{\theta \in \Theta} \mathbb{E}_{\mathbf{X} \sim \mathcal{D}_{\mathcal{X}}} \left[ \Delta_{\mathcal{V}}^{\mathsf{uni}}(g_\theta(\mathbf{X}), f, \mathbf{X}, \alpha) + \lambda_1 \cdot ||g_\theta(\mathbf{X})||_1 + \lambda_{\mathrm{TV}} \cdot ||g_\theta(\mathbf{X})||_{\mathrm{TV}} \right] \tag{56}$$

To learn sufficient and necessary explainer models, we solve Eq. (8) via empirical risk minimization for $\alpha \in \{0, 1\}$ respectively. Given $N$ samples $\{\mathbf{x}_i\}_{i=1}^{N} \overset{\text{i.i.d.}}{\sim} \mathcal{D}_X$, we solve

$$\frac{1}{N} \sum_{i=1}^{N} \left[ \Delta_{\mathcal{V}}^{\mathsf{uni}}(g_\theta(\mathbf{x}_i), f, \mathbf{x}_i, \alpha) + \lambda_1 \cdot ||g_\theta(\mathbf{x}_i)||_1 + \lambda_{\mathrm{TV}} \cdot ||g_\theta(\mathbf{x}_i)||_{\mathrm{TV}} \right]. \tag{57}$$

Here

$$\Delta_{\mathcal{V}}^{\mathsf{uni}}(g_\theta(\mathbf{x}_i), f, \mathbf{x}_i, \alpha) = \alpha \cdot |f(\mathbf{x}_i) - f_S(\mathbf{x}_i)| + (1 - \alpha) \cdot |f(\mathbf{x}_i) - f_S(\mathbf{x}_i)| \tag{58}$$

where $f_S(\mathbf{x}_i)$ is evaluated in the same manner as in the RSNA experiment. For $\alpha = 0$, $\lambda_1 = 0.1$ and $\lambda_{\mathsf{TV}} = 100$. For $\alpha = 1$, $\lambda_1 = 1$ and $\lambda_{\mathsf{TV}} = 10$. For both $\alpha$, during optimization we use a batch size of 32, set $K = 10$ and use the Adam optimizer with default $\beta$-parameters of $\beta_1 = 0.9$, $\beta_2 = 0.99$ and a fixed learning rate of $1 \times 10^{-4}$

**Sampling.** To generate the samples in Figs. 5, 6, 13 and 14, we use the `CoPaint` method [46]. We utilize their code base and pretrained diffusion models (available at `https://github.com/UCSB-NLP-Chang/CoPaint`) with the exact the same parameters as reported in the paper to perform conditional generation.

## A.4. Additional Results

### A.4.1. Natural Language Sentiment Classification

**Analysis of Post-hoc Methods.**

Table 4: Jaccard Index between the sets generated by Integrated Gradients and GradientShap and the optimal solutions $S_{\mathsf{suf}}^*$, $S_{\mathsf{nec}}^*$, and $S_{\mathsf{uni}}^*$ for tweets from the SemEval-2017 dataset.

|  | $\rho = 0.05$ | | $\rho = 0.10$ | | $\rho = 0.15$ | |
| --- | --- | --- | --- | --- | --- | --- |
|  | IG | GS | IG | GS | IG | GS |
| $J(\hat{S}, S_{\mathsf{suf}}^*)$ | $0.72 \pm 0.07$ | $0.61 \pm 0.09$ | $0.73 \pm 0.06$ | $0.64 \pm 0.08$ | $0.67 \pm 0.05$ | $0.59 \pm 0.06$ |
| $J(\hat{S}, S_{\mathsf{nec}}^*)$ | $0.74 \pm 0.07$ | $0.65 \pm 0.09$ | $0.69 \pm 0.06$ | $0.63 \pm 0.08$ | $0.63 \pm 0.06$ | $0.59 \pm 0.06$ |
| $J(\hat{S}, S_{\mathsf{uni}}^*)$ | $0.73 \pm 0.07$ | $0.62 \pm 0.09$ | $0.77 \pm 0.07$ | $0.69 \pm 0.08$ | $0.71 \pm 0.05$ | $0.64 \pm 0.06$ |

Table 5: Jaccard Index between the sets generated by Integrated Gradients and GradientShap and the optimal solutions $S_{\mathsf{suf}}^*$, $S_{\mathsf{nec}}^*$, and $S_{\mathsf{uni}}^*$ for tweets from the SemEval-2017 dataset.

|  | $\rho = 0.35$ | | $\rho = 0.40$ | | $\rho = 0.45$ | |
| --- | --- | --- | --- | --- | --- | --- |
|  | IG | GS | IG | GS | IG | GS |
| $J(\hat{S}, S_{\mathsf{suf}}^*)$ | $0.58 \pm 0.03$ | $0.55 \pm 0.03$ | $0.58 \pm 0.04$ | $0.54 \pm 0.03$ | $0.60 \pm 0.04$ | $0.56 \pm 0.04$ |
| $J(\hat{S}, S_{\mathsf{nec}}^*)$ | $0.50 \pm 0.04$ | $0.50 \pm 0.04$ | $0.51 \pm 0.03$ | $0.52 \pm 0.04$ | $0.51 \pm 0.03$ | $0.51 \pm 0.04$ |
| $J(\hat{S}, S_{\mathsf{uni}}^*)$ | $0.56 \pm 0.04$ | $0.53 \pm 0.03$ | $0.56 \pm 0.04$ | $0.52 \pm 0.03$ | $0.55 \pm 0.03$ | $0.55 \pm 0.03$ |

**Sufficiency vs Necessity**

Table 6: Comparison of solutions $S_{\mathsf{suf}}^*$, $S_{\mathsf{nec}}^*$, and $S_{\mathsf{uni}}^*$ on the SemEval-2017 dataset.

|  | $\rho = 0.05$ | $\rho = 0.10$ | $\rho = 0.15$ | $\rho = 0.40$ | $\rho = 0.45$ |
| --- | --- | --- | --- | --- | --- |
| $J(S_{\mathsf{suf}}^*, S_{\mathsf{nec}}^*)$ | $0.85 \pm 0.06$ | $0.72 \pm 0.06$ | $0.59 \pm 0.05$ | $0.56 \pm 0.04$ | $0.54 \pm 0.03$ |
| $J(S_{\mathsf{suf}}^*, S_{\mathsf{uni}}^*)$ | $0.96 \pm 0.04$ | $0.83 \pm 0.05$ | $0.73 \pm 0.05$ | $0.63 \pm 0.05$ | $0.64 \pm 0.04$ |
| $J(S_{\mathsf{nec}}^*, S_{\mathsf{uni}}^*)$ | $0.88 \pm 0.06$ | $0.85 \pm 0.06$ | $0.78 \pm 0.07$ | $0.65 \pm 0.06$ | $0.63 \pm 0.04$ |

**Example Solutions to** $(P_{suf})$**,** $(P_{nec})$**, and** $(P_{uni})$

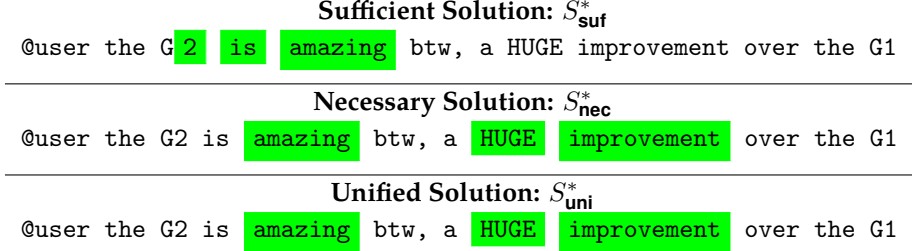

Figure 9: Solutions ($\rho = 0.15$), $S^*_{suf}$, $S^*_{nec}$, and $S^*_{uni}$, for a tweet from the SemEval-2017 dataset.

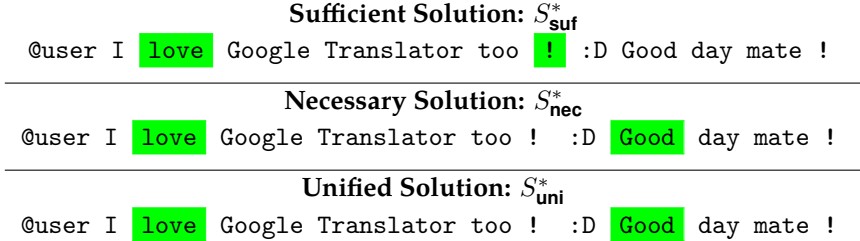

Figure 10: Solutions ($\rho = 0.10$), $S^*_{suf}$, $S^*_{nec}$, and $S^*_{uni}$, for a tweet from the SemEval-2017 dataset.

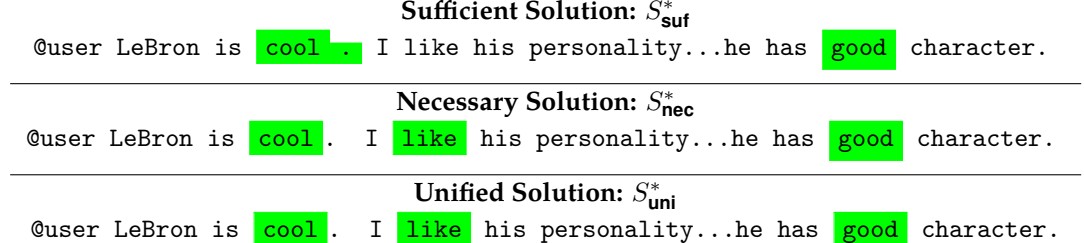

Figure 11: Solutions ($\rho = 0.15$), $S^*_{suf}$, $S^*_{nec}$, and $S^*_{uni}$, for a tweet from the SemEval-2017 dataset.

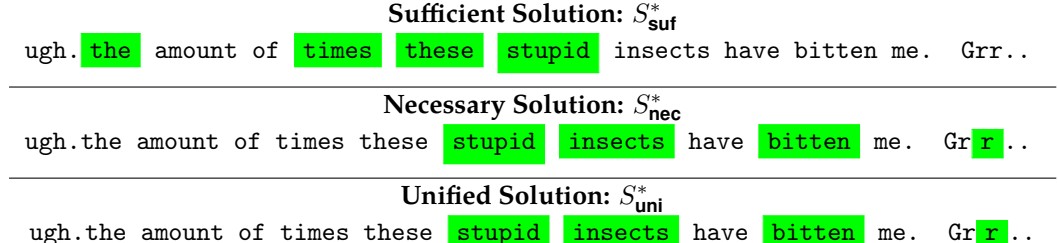

Figure 12: Solutions ($\rho = 0.25$), $S^*_{suf}$, $S^*_{nec}$, and $S^*_{uni}$, for a tweet from the SemEval-2017 dataset.

### A.4.2. CelebA-HQ

**Keeping and removing the sufficient subset $S^*_{\text{suf}}$**

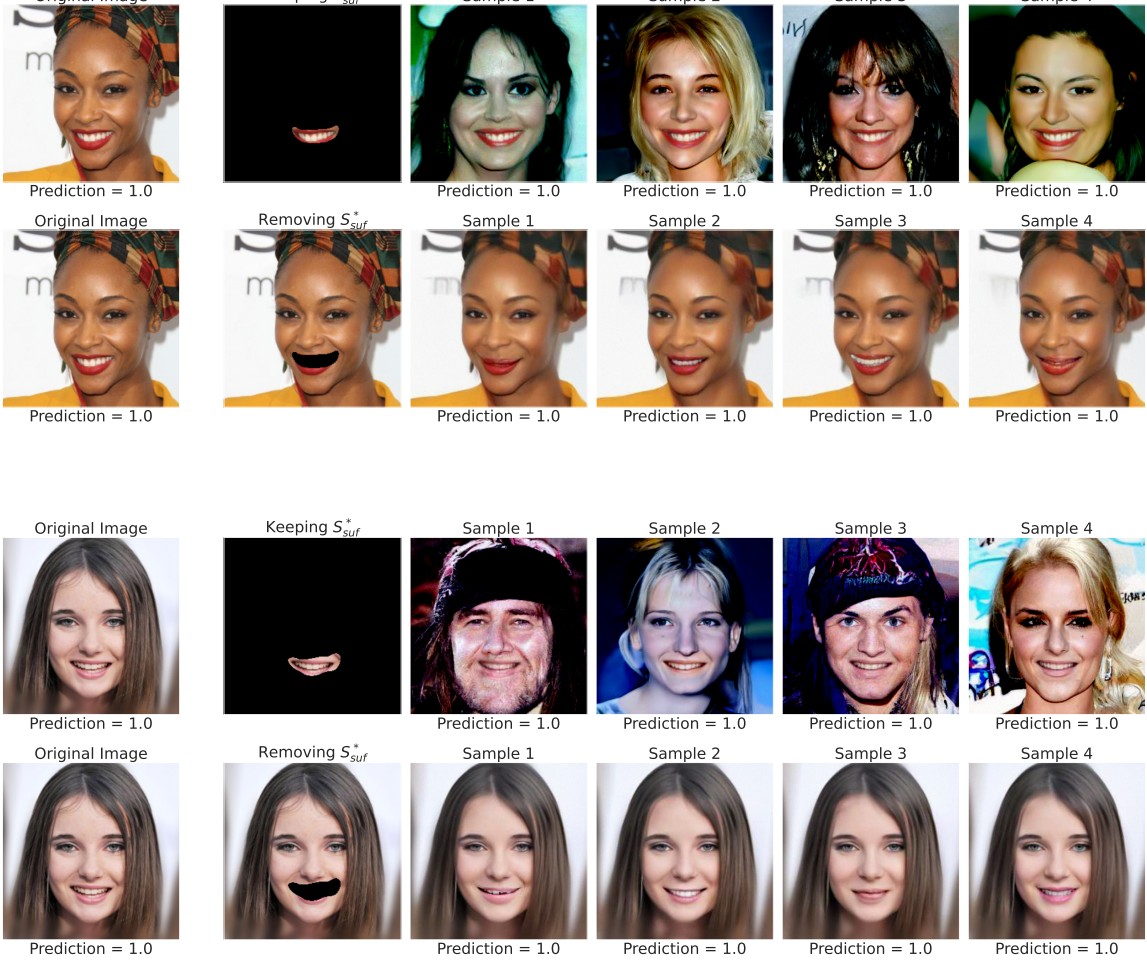

Figure 13: Images and model predictions by keeping and removing the sufficient subset $S^*_{\text{suf}}$.

**Keeping and removing the necessary subset $S^*_{\textsf{nec}}$**

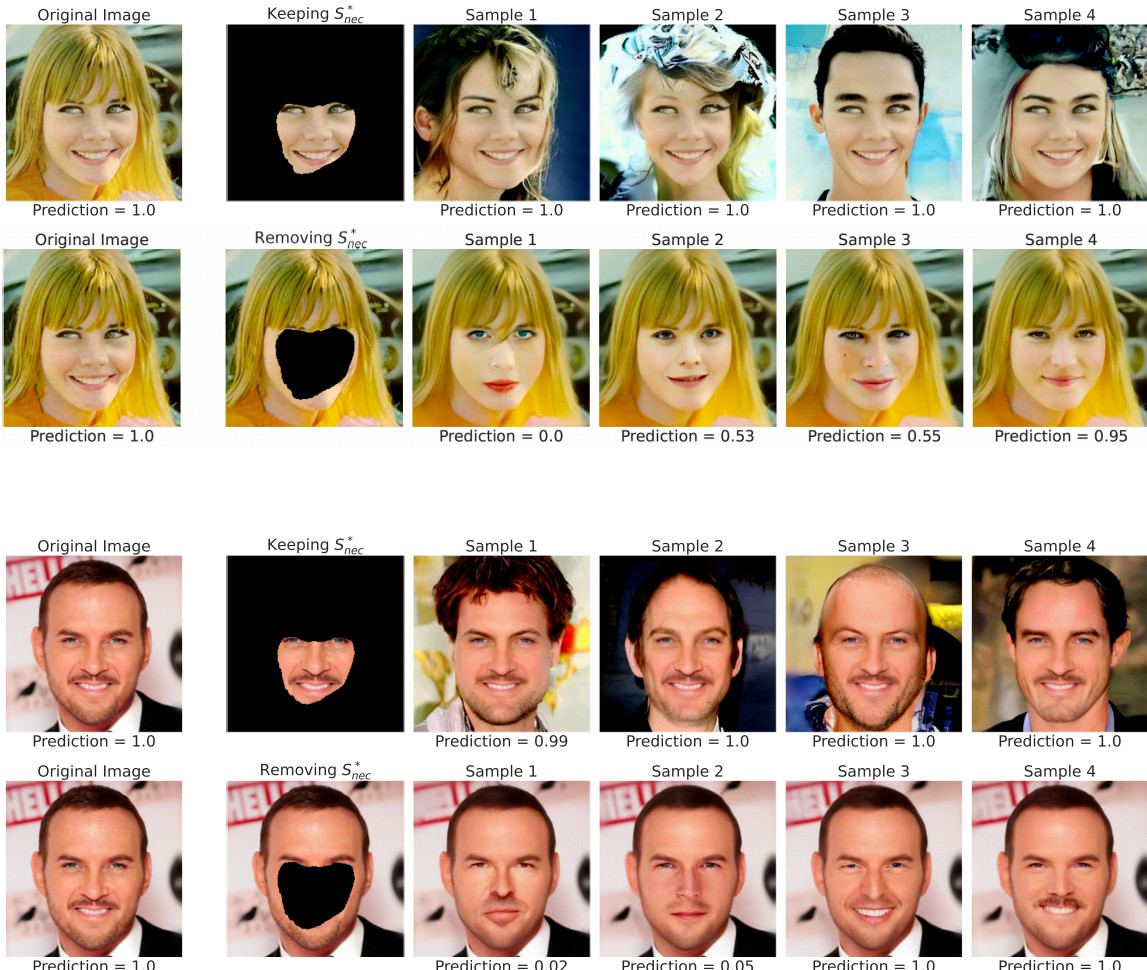

Figure 14: Images and model predictions by keeping and removing the necessary subset $S^*_{\textsf{nec}}$.

