# OpenReview forum: "Sufficient and Necessary Explanations (and What Lies in Between)"
_CPAL.cc/2025/Proceedings_Track — CPAL 2025 (Proceedings Track) Oral_

### Official Review · Reviewer_xkYE · 2025-01-07

**Rating:** 6
**Confidence:** 3

**Review:**

This paper provides a formal approach to evaluating machine learning model explanations through the concepts of sufficiency and necessity. It challenges the effectiveness of current post-hoc explanation methods by demonstrating that many commonly used techniques tend to offer explanations that are only sufficient but not necessary. By developing a unified framework that spans the necessity-sufficiency axis, the authors propose a method that provides more comprehensive insights into feature importance and aligns explanations.

Pros:

1.	The paper introduces a novel framework that combines the concepts of sufficiency and necessity in explanations, offering a more nuanced understanding of feature importance.

2.	It establishes formal mathematical definitions and leverages them to critically assess the limitations of existing explanation methods.

3.	The approach is validated with extensive experiments that illustrate how the new method can reveal important features that traditional methods might miss.

Cons:

1.	In section 6, there is a need for further discussion on the applicability of the framework across different types of machine learning models and settings, as well as its adaptability to various real-world scenarios where explainability is crucial.

---

### Official Review · Reviewer_sT6t · 2025-01-14
**Solid paper and insightful theoretical results**

**Rating:** 8
**Confidence:** 2

**Review:**

Summary:

The paper makes a significant contribution to explainable AI by formalizing and unifying sufficiency and necessity concepts. The framework is validated on both synthetic and real-world scenarios.

Strength:
- The paper introduces a unified framework that bridges sufficiency and necessity in explaining model predictions. This novel approach is well-motivated by the limitations of existing post-hoc interpretability methods.
- Precise mathematical formulations of sufficiency and necessity are provided.
- The connection to Shapley values and the game-theoretic interpretation of the unified framework provide a novel angle to quantify feature importance.

---

### Official Review · Reviewer_Mh2W · 2025-01-17

**Rating:** 7
**Confidence:** 2

**Review:**

This paper explores the interplay of sufficiency and necessity in determining feature importance. The authors both theoretical and empirical justifications on their insight.

Pros:
- Math is accompanied by clear explanations in plain English
- Intuitive experiments

Cons:
- I'm a bit unclear on what the potential downstream applications of these findings are. It would be good to emphasize this to motivate your work

Questions:
1) How do your results carry over to more complex features? In your experiments, you primarily look at the importance of each pixel, but what happens if you were not interested in the underlying image but a (possibly nonlinearly) transformed version of it?
2) Can you transfer your results to discrete data like in NLP?
3) Have you tried noisy images?

---

### Meta-Review · Area_Chair_wbRZ · 2025-02-04

**Recommendation:** Accept (Oral)
**Confidence:** 4

**Metareview:**

This paper addresses the challenge of understanding complex machine learning (ML) models by formalizing the concepts of sufficiency and necessity in feature importance explanations. The authors argue that traditional post-hoc explanation methods often provide ambiguous insights into which features are crucial for a model's predictions. To improve clarity, they introduce precise mathematical definitions for sufficient and necessary features and propose a unified framework to spans the necessity-sufficiency axis, showing how this unified notion relates to established feature importance measures like conditional independence and Shapley values.

### Pros
* The formalization of sufficiency and necessity provides a rigorous framework for understanding feature importance in ML models。
* The paper presents clear mathematical definitions and principles, enhancing the theoretical foundation of feature importance explanations.
* The authors conduct a range of experiments that effectively illustrate the utility of their proposed framework, showcasing how it can uncover critical insights into model behavior.

### Cons
* The paper could benefit from more extensive empirical validation across diverse datasets and models to demonstrate the practical applicability of the proposed methods.

The consensus among reviewers was that the paper provides a fresh and solid contribution to the field of understanding machine learning models, and unanimously recommended acceptance.

**Additional Comments On Reviewer Discussion:**
* Reviewer Mh2W inquired about the downstream applications of the findings, which the authors addressed sufficiently.
* Reviewer xkYE expressed a desire for the authors to consider a more comprehensive evaluation of the experiments across different types of machine learning models and settings. The authors explain about the future plan according to that.

---

### Decision · Program_Chairs · 2025-02-11

Accept (Oral)